# Advanced Manufacturing in the Fabrication of a Lifelike Brain Glioblastoma Simulator for the Training of Neurosurgeons

**DOI:** 10.3390/polym14061072

**Published:** 2022-03-08

**Authors:** Pin-Chuan Chen, Yu-Wen Yang, Jang-Chun Lin, Wei-Hsiu Liu

**Affiliations:** 1Department of Mechanical Engineering, National Taiwan University of Science and Technology, Taipei 106, Taiwan; pcchen@mail.ntust.edu.tw (P.-C.C.); lylidy65@gmail.com (Y.-W.Y.); 2High Speed 3D Printing Research Center, National Taiwan University of Science and Technology, Taipei 106, Taiwan; 3Department of Radiation Oncology, Shuang Ho Hospital, Taipei Medical University, Taipei 106, Taiwan; 13451@s.tmu.edu.tw; 4Department of Radiology, School of Medicine, College of Medicine, Taipei Medical University, Taipei 106, Taiwan; 5Department of Neurological Surgery, Tri-Service General Hospital and National Defense Medical Center, Taipei 114, Taiwan; 6Department of Surgery, School of Medicine, National Defense Medical Center, Taipei 114, Taiwan

**Keywords:** neurosurgeon surgical simulator, brain simulator, cerebral cancer surgery practice

## Abstract

Neurosurgeons require considerable expertise and practical experience to deal with the critical situations commonly encountered in complex surgical operations such as cerebral cancer; however, trainees in neurosurgery seldom have the opportunity to develop these skills in the operating room. Physical simulators can give trainees the experience they require. In this study, we adopted advanced molding and replication techniques in the fabrication of a physical simulator for use in practicing the removal of cerebral tumors. Our combination of additive manufacturing and molding technology with elastic material casting made it possible to create a simulator that realistically mimics the skull, brain stem, soft brain lobes, and cerebral cancer with cerebral tumors located precisely where they are likely to appear. Multiple and systematic experiments were conducted to prove that the elastic material used herein was appropriated for building professional medical physical simulator. One neurosurgical trainee reported that under the guidance of a senior neurosurgeon, the physical simulator helped to elucidate the overall process of cerebral cancer removal and provided a realistic impression of the tactile feelings involved in craniotomy. The trainee also learned how to make decisions when facing the infiltration of a cerebral tumor into normal brain lobes. Our results demonstrate the efficacy of the proposed physical simulator in preparing trainees for the rigors involved in performing highly delicate surgical operations.

## 1. Introduction

Formations of cancer cells in the brain (i.e., tumors) can be categorized as primary (starting within the brain) and secondary (spreading from tumors located outside the brain), otherwise known as brain metastasis. Tumors are generally classified as malignant or benign; however, differentiating between the two can be exceedingly difficult, such that all formations must be treated as potentially dangerous. One common form of brain tumor is referred to as glioblastoma, a malignant tumor centered in the deep white matter of the cerebral hemispheres, most frequently in the frontal lobe.

Neurosurgeons require extensive knowledge and well-trained surgical skills to deal effectively with cerebral tumors. They also require practical experience to defuse the crises encountered during difficult surgical operations [1]. Under the guidance of instructors, cadavers can be used to familiarize students with the structure of the brain. Unfortunately, cadavers are costly and difficult to obtain, they do not present an accurate representation of the blood flow or other physical phenomena encountered in live patients, and many students are bothered by the smell of formaldehyde used in embalming. Furthermore, hospital administrators must ensure the safety of patients by limiting access to all but the most highly trained personnel, such that many junior neurosurgeons have limited opportunities for practical training [1,2]. This situation has prompted the use of medical simulators as a proxy for practical experience. In 2011, the Association of American Medical Colleges in 2011 issued the following statement: “Simulation has the potential to revolutionize healthcare and address patient safety issues if appropriated used and integrated into the educational and organizational improvement process” [3,4,5].

Virtual reality (VR) devices are increasingly being used to develop computer-based models as teaching aids for medical students [6,7,8,9,10] and rapid prototyping based on 3D printing technology is being used to create physical devices for the same purpose [11,12,13,14,15,16,17,18]. VR technology was first used for the training of surgeons by Lanier in 1987. In another early study, Delp et al. [9] developed a VR simulation for the repair of the Achilles’ tendon. Since that time, a plethora of computer-based simulations have been developed. Pelargos et al. [10] reviewed the use of VR and augmented reality (AR) in the training of neurosurgeons, including the history, current status, future potential, and limitations. VR and AR can be used to learn patient-specific anatomy, select appropriate surgical instruments, plan responses to complications, and enhance operational efficiency. Enabling neurosurgeons to learn and rehearse surgical procedures reduces the likelihood of error. Choudhury et al. [7] were the first to use a VR simulator to standardize training for neurosurgical oncology. Experimental results demonstrated the efficacy of this approach in preparing neurosurgical residents through the development of technical skills. However, virtual reality medical simulators are expensive, they do not provide the tactile feedback of actual surgeries, and the images do not provide an accurate representation of actual anatomic features.

Recent improvements in 3D printing techniques have greatly improved the fidelity of physical models used in practice surgeries, and expanded the availability of simulators for research and training. D’urso et al. [11] and Wurm et al. [18] fabricated cerebrovascular simulators using stereolithography (SL) to create aneurysms and the surrounding blood vessels. Their devices provide an overview of relevant anatomic structures to assist in planning surgical procedures, such as the selection of appropriate aneurysm clips. Wurm et al. [17] used SL in conjunction with 3D printing to create a solid aneurysm as well as the surrounding vessels and neurocranium. They described 3D printing as the rapid prototyping technique with the greatest promise for the creation of neurovascular structures. Kimura et al. [13] used 3D printing to create a hollow 3D model of semi-elastic aneurysms. Their model is meant to provide practice in performing craniotomy, which re- quires drilling into the base of the skull and subsequent clipping. They also fabricated an aneurysm with surrounding vessels and cranial base bones mimicking those encountered in difficult clinical cases. Mashihiro et al. [14] employed 3D printing to create a solid aneurysm using acrylonitrile butadiene styrene (ABS), which was then coated with pink liquid silicone. The subsequent use of xylene to dissolve the ABS left a hollow 3D device with the elastic properties required to simulate the clipping of aneurysms.

Ploch et al. [19] used a nondeformable 3D-printed brain model as a template for molding and casting a soft deformable material referred to as synthetic gelatin. The efficacy of the model was demonstrated in a multicenter survey of neurosurgeons and surgical residents in Europe and the United States. Qing et al. [20] used 3D printing technology to manufacture a model for practice in performing keyhole surgical operations on complex intracranial lesions, such as brain tumors and aneurysms. Note, however, that the 3D-printing materials used in their model did not mimic the density, compressibility, or elasticity of actual brain tissue, brain tumors, and aneurysms. Chanda et al. [21] developed an inexpensive two-part silicone-based casting system mimicking the nonlinear mechanical properties of white and grey matter. Their novel brain tissue surrogates based on five hyperelastic material models can be used for the study of TBI as well as training in a clinical setting. Mussi et al. [22] produced a low-cost 3D patient-specific anatomical model using 3D printing in conjunction with materials that accurately mimic real tissue. Four case studies demonstrated the efficacy of their model in preoperative planning. Coelho et al. [23] employed 3D printing in the fabrication of a highly accurate frontoethmoidal meningoencephalocele model, which mimics the consistency and resistance of multiple tissue types. Their 3D-printed model is meant to facilitate the planning of osteotomies and the pre-contouring of osteosynthetic material to enhance surgical efficiency.

In the current study, we employed additive manufacturing to create highly detailed molds of the brain stem, soft brain lobes, skull, and cerebral cancer embedded within normal brain tissue. Multiple experiments were conducted to characterize the material properties of the physical simulators, while a senior neurosurgeon confirmed that the proposed medical simulator accurately mimics the tactile feeling when dealing with real tissue. Overall, the proposed simulator proved highly effective in training neurosurgical trainees to perform a variety of procedures from craniotomy to tumor removal.

## 2. Fabrication of Proposed Medical Simulator

Our primary objective in this study was to create a lifelike cerebral simulator for training neurosurgery students. Additive manufacturing and multiple casting processes were used to create a highly complex medical simulator, which included the skull, brain stem, soft brain lobes, and a cerebral cancer deeply embedded within a normal brain. Our initial efforts to mimic the tactility of the various structures was hindered by a lack of data pertaining to the material properties of cerebral cancer and brain tissue. We therefore relied on the experience of a senior neurosurgeon in assessing the models. An industrial 3D scanner (Artec Space Spider, Artec3D, Luxembourg) was used to scan the physical model used at Tri-Service General Hospital, whereupon the digital file was converted into STL file format for additive manufacturing. The location, size, and morphology of the cerebral cancer were based on consultations with a senior neurosurgeon and oncologist and revised from the medical image of a patient who was a 67-year-old female with pathology of glioblastoma.

### 2.1. Manufacturing of Skull and Brain Stem

An FDM 3D printer (Fortus 360mc, Stratasys, Eden Prairie, MN, USA) and stereolithographic (SL) printer (Form 3, formlabs, Somerville, MA, USA) were used to print the skull and brain stem using acrylonitrile butadiene styrene (ABS). Figure 1a illustrates the 3D-printed skull, Figure 1b illustrates the 3D-printed brain stem, and Figure 1c illustrates the assembled skull and brain stem. Please note that the design and positioning of the pins and hooks in the top skull, bottom skull, and the brain stem (Figure 1b,c) were meant to facilitate rapid assembly.

### 2.2. Fabrication of Brain Lobes and Cerebral Tumor

The consulting neurosurgeon clarified the important differences between cerebral cancer tissue and normal brain tissue in terms of tactility and color. We employed the similar molding methods for cancer tissue and brain lobes; however, we adopted different methods for the preparation of the respective materials. Figure 2 illustrates the process used in fabricating the brain lobes. We began by 3D printing (Fortus 360 mc, Stratasys, Eden Prairie, MN, USA) an ABS mold (ABS 1.75 mm, White, Taipei, Taiwan) (see Figure 2a). The printing condition are layer height of 1.8 mm and print speed of 60 mm/s. The mold was fixed within a transparent box (see Figure 2b), into which was poured white silicon to create a cast (see Figure 2c). The cast was removed from the box and sliced open using a knife in order to remove the mold from within (see Figure 2d). Into the resulting cavity, we poured a mixture of glycerin, jelly wax, and thermoplastic rubber at a specific ratio, which was dyed beige (rather than the gray color used for tumors) (see Figure 2e). The resulting model of a brain lobe is shown in Figure 2f. Please note that tumors feel slightly softer than normal brain tissue; however, they are also somehow less flexible. Thus, we used roughly the similar method to create the tumors, but the jelly mixture included only glycerin and jelly wax to produce a slightly flaccid structure.

### 2.3. Fabrication of Brain with Cerebral Tumor

The brain lobes were fabricated in four pieces: frontal lobe, parietal lobe, temporal lobe, and occipital lobe, as shown in Appendix A. We embedded a cerebral tumor within the right parietal lobe using the same basic methods shown in Figure 2; however, the entire lobe was not cast all at once. Rather, we poured the jelly mixture only into one half of the white silicon cast shown in Figure 3a, into which we placed the preformed tumor (see Figure 3b). We then closed the white silicon cast and poured in the remaining jelly mixture (see Figure 3c). The brain lobe with embedded cerebral tumor is shown in Figure 3d.

### 2.4. Assembly of Medical Simulator

The brain stem and brain lobes were assembled within a preformed skull. Figure 4a presents a top-view image of the frontal lobe and parietal lobe with the top skull removed, Figure 4b presents a side-view image of the complete simulator with the top skull, frontal lobe, and parietal lobe removed and Figure 4c shows the entire simulator ready for surgery practice. 

## 3. Material Testing and Analysis

Since the tactile feeling of the physical simulator is critical to the training of trainees, and limited information on real brains can be found from the previous literature, the tactile feeling of the physical simulator was confirmed by a senior neurosurgeon. To further understand the material properties of the physical simulator, especially the brain lobes, multiple experiments were conducted. Two types of mixture materials were studied herein, including the jelly wax only and jelly wax with thermoplastic rubber, and the aim of studying materials was to ensure that the properties of the brain lobes were as similar as the real brain during the surgery process. 

### 3.1. The Collapse Phenomenon of the Brain Lobe

Maintaining the original shape of brain lobe is one of the important characteristics for the physical simulator; therefore, experiments were designed to compare the collapse phenomenon between jelly wax only and jelly wax with thermoplastic rubber. From our observation, the volume of the simulator did not change, but the height or width would change because of deformation instead of evaporation, because no water was involved in the material. In Appendix A, it can clearly be seen that the deformation (in terms of height and width) of the jelly wax plus thermoplastic rubber is smaller than that of the jelly wax. The change in height is more significant than the change in width; therefore, height was used as an indicator to demonstrate the deformation of the simulator in terms of time. In the experiments, both materials were used to cast brain lobes and the height of the brain lobes was measured every day for two weeks. Figure 5a,b show cross-sectional views of brain lobe, where Figure 5a was cast from jelly wax only, while Figure 5b was cast from Jelly wax and thermoplastic rubber. The transparency difference between Figure 5a,b was a result of the thermoplastic rubber. Figure 5a shows the measured height of the brain lobe made of jelly wax after 14 days, Figure 5b shows the measured height of brain lobe made of jelly wax and thermoplastic rubber after 14 days, and Figure 5c shows the measured height of brain lobes from day zero to day fourteen for both cases. The initial height of the brain lobe made of jelly wax was 24.76 mm, while the initial height of brain lobe made of jelly wax and thermoplastic rubber was 25.73 mm. After two weeks, it is obvious that the height of the brain lobe made of jelly wax decreased much faster than the height of the brain lobe made of jelly wax and thermoplastic rubber, resulting the final height of the brain lobe made of jelly wax was16.18 mm (65% of the initial height) and the final height of the brain lobe made of jelly wax and thermoplastic rubber was 20.65 mm (80% of the initial height). In other words, the brain lobe made of jelly wax and thermoplastic rubber maintained its original shape much longer, which is more suitable for being used for physical simulators. 

### 3.2. The Recovery Performance of the Brain Lobe

According to the experience of sensor neurosurgeon doctors, human brain lobes have the capability to heal themselves after surgery. Therefore, the space created between brain lobes during the surgery process by using surgical forceps recovers by itself. Therefore, experiments were conducted to understand the recovery performance of brain lobes made of two materials: jelly wax only and jelly wax with thermoplastic rubber. In the experiments, hemispheric objects were cast with both materials, a cut was created on the top of the hemispheric object, and a V-clamp was inserted into the cut for 4 h or 8 h, representing surgical forceps being used to make space between the brain lobes during surgery (Figure 6a) for 4 h or 8 h. Figure 6b shows the experimental results of jelly wax after 8 h, while Figure 6c shows the experiment result of jelly wax and thermoplastic rubber after 8 h; obviously, there is a crack in Figure 6b, showing that the jelly wax was not able to completely recover. To further understand the recovery phenomenon, the width of the cut on the hemispheric object was measured several times right after the V-clamp was removed from the hemispheric object for either 4 h or 8 h (representing surgery operations lasting for either 4 h or 8 h). Figure 6d shows the recovery ratio after 4 h and Figure 6e shows the recovery ratio after 8 h. It is clear that: (1) the brain lobe made of jelly wax and thermoplastic rubber was able to completely recover, while the brain lobe made of jelly wax only was only partially able to recover; and (2) for the brain lobe made of jelly wax only, the duration for which the V-clamp was applied (either 4 h or 8 h) influenced the recovery—the longer the V-clamp was inserted in the brain lobes, the lower the achieved recovery ratio (the recovery ratio was 83% for 4 h, while the recovery ratio was 67% for 8 h). In summary, if the brain lobe is made of jelly wax and thermoplastic rubber, then the brain lobe can be fully recovered after removal of the surgical forceps from the brain lobes, which better reflects the real situation.

### 3.3. Tension Test

During the training with the physical simulator, the brain lobes will be frequently touched by trainees with different levels of force; therefore, a tension test was performed to understand the maximum loading that both materials could survive. A tensile testing machine (Instron 3365, Norwood, MA, USA) was used to conduct the experiment of the brain lobes made of either jelly wax only or jelly wax and thermoplastic rubber. Conventionally, tension testing can be used to determine the Young’s modulus, yielding stress, and maximum stress on the basis of the stress–strain curve, but it is quite difficult to quantify the materials used in this research (the experimental results are shown in the Appendix A). Therefore, a maximum tensile test was used to demonstrate the material difference between jelly wax only and jelly wax plus thermoplastic rubber. For each material, ten specimens were fabricated and tested, and the results are shown in Figure 7. It is clear that the average maximum loading of the jelly wax was 1.8 × 10^−4^ kN, while the average maximum loading of the jelly wax and thermoplastic rubber was 2.9 × 10^−4^ kN. In other words, the brain lobes made of jelly wax and thermoplastic rubber would be more durable for use in physical simulators.

## 4. Practicing Surgical Techniques

Figure 8a shows an experienced neurosurgeon (Dr. Liu from our research team) giving a neurosurgery trainee instructions on the removal of a cerebral tumor using the proposed simulator. Please note that the entire procedure is detailed in a Appendix A (26 February 2022 https://youtu.be/49MQlXRNBBQ). The fact that the cerebral tumor was located in the right parietal lobe necessitated adjustment of the head position to a specific angle and orientation prior to commencing surgery (Figure 8b). Before initiating the craniotomy, three blue marks were painted on the skull as guides in sawing the top skull (Figure 8c). A surgical saw was used to cut off the top skull to expose the brain lobes (Figure 8d,e). A surgical retractor was used to separate the brain lobes in order to locate the tumor (Figure 8f). After the tumor was positively identified, surgical forceps were used to remove the via multiple scooping actions, under the guidance of Dr. Liu. Removal of the grey tumor material left a cavity (Figure 8g,h). Note the completeness and fidelity of the simulator, which even included corpus callosum (Figure 8h). The forceps were withdrawn, and the skull covered to complete the operation (Figure 8i).

The senior neurosurgeon and trainee were both very impressed with the simulator. They were particularly pleased with the tactile feeling of the skull and the fact that the tumor was hidden within normal brain lobes. The greatest challenge in the treatment of cerebral cancer is achieving complete tumor removal without damaging normal brain structures. It is crucial that neurosurgery trainees develop an understanding of the point at which the removal of tissue should be halted. However, this can be exceedingly difficult in many situations where the boundary between tumor tissue and normal brain tissue is not clearly defined. As shown in Figure 8h, the proposed simulator created just such a situation in which tumor tissue (grey color) was infiltrated with normal tissue (beige-color). This is precisely the kind of situation in which clinicians must decide whether to proceed with excision or leave the unremoved cancer to chemotherapy. Note also that the skull we created in this study can be separated into three layers (outside cortex, middle marrow, and inside cortex), the marrow of which is the most fragile. Our novel use of 3D printing to manufacture hollow structures proved highly effective in mimicking the middle marrow layer sandwiched between two solid layers (outside and inside cortex). The overall effect was highly accurate representation of an actual cranium.

During an interview, the trainee reported that the physical simulator gave him a deeper understanding of the overall tumor removal process, including the tactile sensations crucial to the success of delicate operations. The trainee also described experiencing a strong emotional reaction when he began removing the tumor. Overall, it appears that the proposed physical model presented precisely the situations one could expect to encounter in a real-world surgical environment.

## 5. Conclusions

Neurosurgeons require professional knowledge as well as practical experience in performing difficult operations. The conventional approach to training using cadavers is expensive and does not lend itself to repeated practice sessions. In this study, we developed a novel molding–casting scheme by which to create a physical simulator to provide practice in the removal of brain tumors. Multiple experiments were conducted to prove that jelly wax and thermoplastic rubber is a perfect candidate for creating brain lobes, because it can minimize the material collapse, improve the durability, and enhance the tensile strength. From the experimental results shown in Figure 5 and Figure 6, the simulator should maintain its functionality after two weeks with slight deformation, but more research will be carried out regarding the maintenance of the original shape for longer periods. The proposed physical simulator included a complete skull, brain stem, soft brain lobes, and tumors located precisely in locations where they are likely to appear. One neurosurgical trainee practicing under the guidance of a senior neurosurgeon reported that the physical simulator clarified the overall process of tumor removal, while providing a realistic impression of the tactile feelings involved in this delicate operation. The three-layer structure of the skull behaves in a manner similar to that of a real skull and the infiltration of the tumor within normal tissue is highly realistic. These results demonstrate the efficacy of the proposed physical simulator in preparing trainees for the rigors involved in performing highly delicate neurological surgical operations. In the near future, we plan to establish professional training courses for medical education organizations with more systematical study, then more feedback can be received from junior and senior neurosurgeons to improve this simulator. 

## Figures and Tables

**Figure 1 polymers-14-01072-f001:**
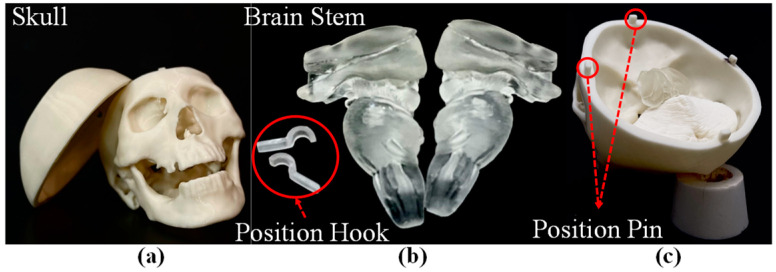
3D-printed parts of medical simulator: (**a**) top and bottom of skull with position pins; (**b**) brain stem with position hooks; (**c**) assembled skull and brain stem.

**Figure 2 polymers-14-01072-f002:**
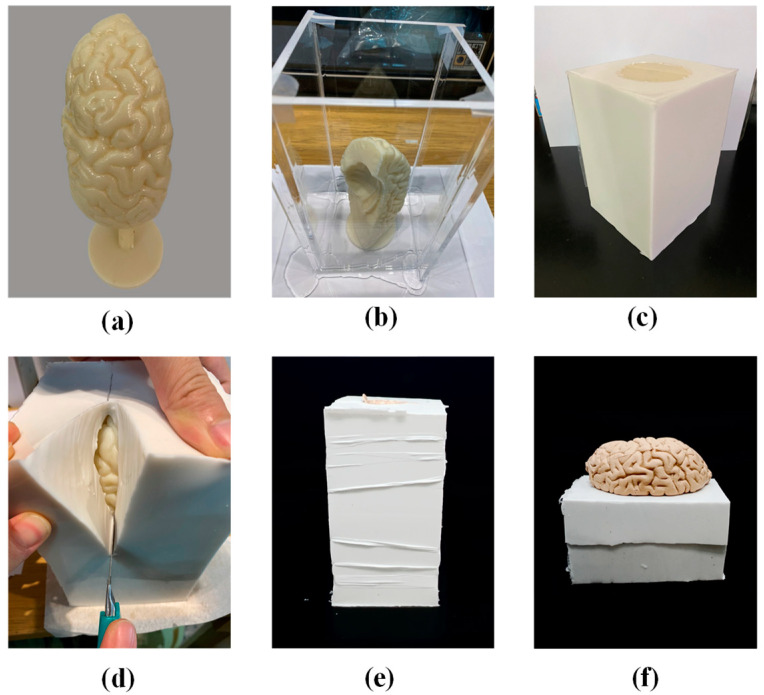
(**a**) Mold of brain lobe printed using a 3D printer (Fortus 360mc, Stratasys) with acrylonitrile butadiene styrene (ABS); (**b**,**c**) the mold was fixed within a transparent box into which white silicon was poured to create a cast; (**d**) the silicon mold was sliced open using a knife and then peeled back to remove the ABS mold from within; (**e**) beige-colored jelly mixture was poured into the cavity; (**f**) to create a realistic model of the brain lobe.

**Figure 3 polymers-14-01072-f003:**
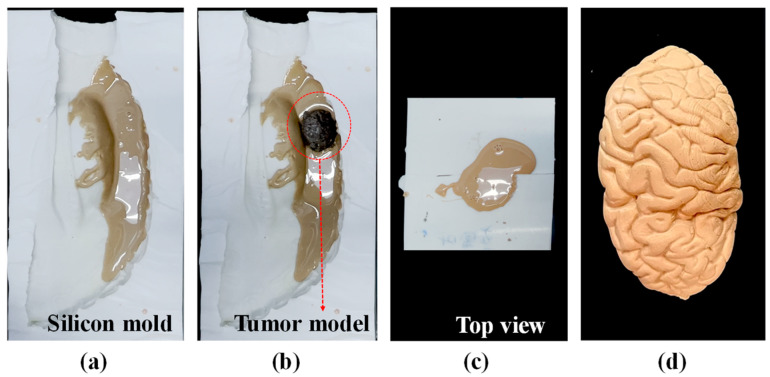
(**a**) A small quantity of jelly was poured into the silicon mold; (**b**) placement of preformed cerebral tumor within the jelly; (**c**) the cast was then closed, into which the remaining jelly mixture was poured; (**d**) finished right parietal lobe with embedded cerebral tumor.

**Figure 4 polymers-14-01072-f004:**
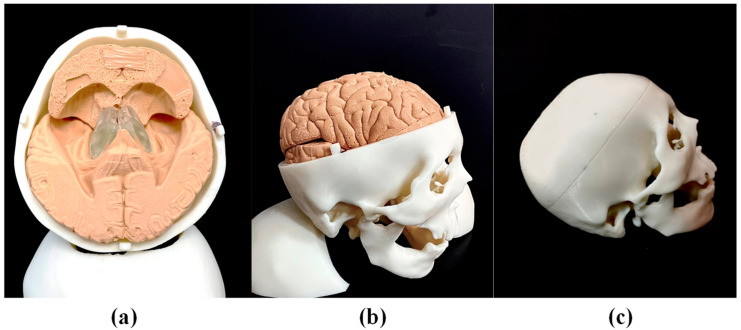
(**a**) Top-view image of simulator prior to assembly of the top skull, frontal lobe, and parietal lobe; (**b**) side-view image of simulator prior to placement of the top skull; (**c**) entire simulator ready for practicing surgical techniques.

**Figure 5 polymers-14-01072-f005:**
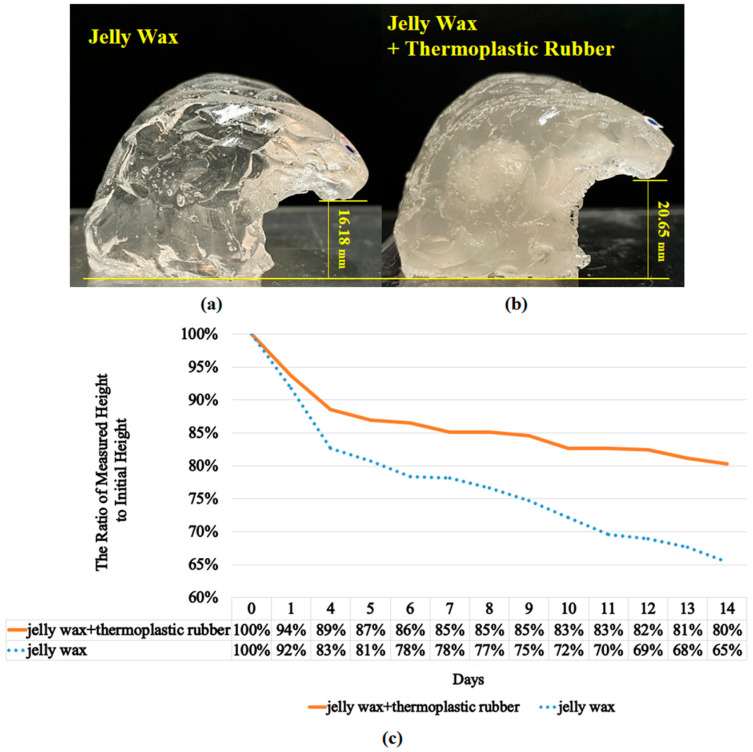
(**a**) Measured height of brain lobe made of jelly wax only after 14 days; (**b**) measured height of brain lobe made of jelly wax and thermoplastic rubber after 14 days; (**c**) comparison chart showing the measured height in terms of days for both cases.

**Figure 6 polymers-14-01072-f006:**
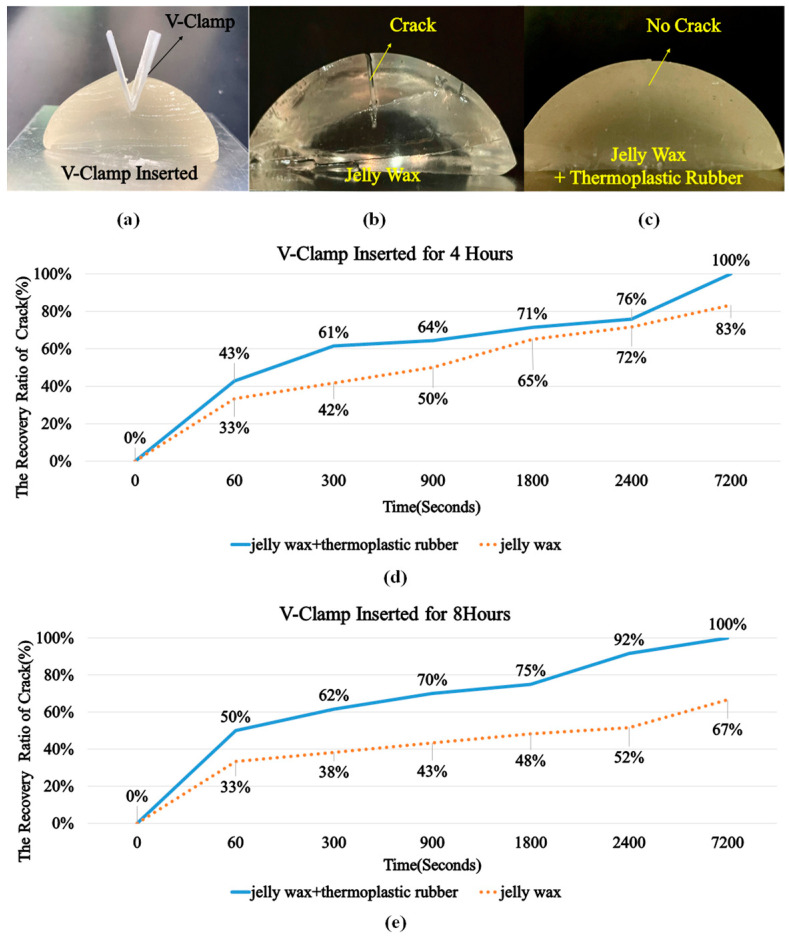
(**a**) A V-clamp was inserted into the cut for either 4 h or 8 h, representing that surgical forceps were used to make space between brain lobes during the surgery for either 4 h or 8 h; (**b**) a crack was observed on the hemispheric object made of jelly wax only; (**c**) no crack was observed on the hemispheric object made of jelly wax and thermoplastic rubber; (**d**) recovery ratio after 4 h for both cases, 100% recovery ratio means no crack was observed and lower recovery ratio means wider crack was found; (**e**) recovery ratio after 8 h for both cases.

**Figure 7 polymers-14-01072-f007:**
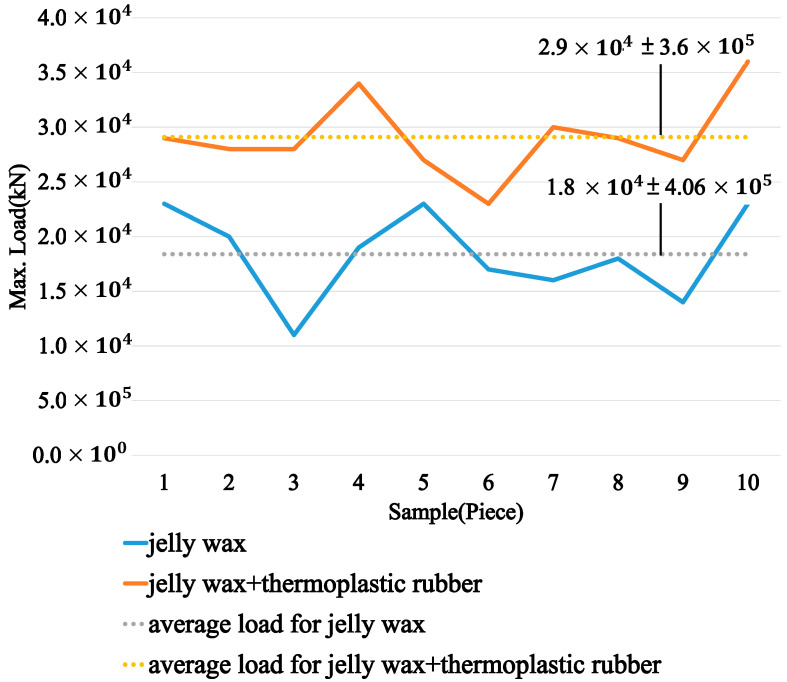
The experimental results of the tensile tests for both materials.

**Figure 8 polymers-14-01072-f008:**
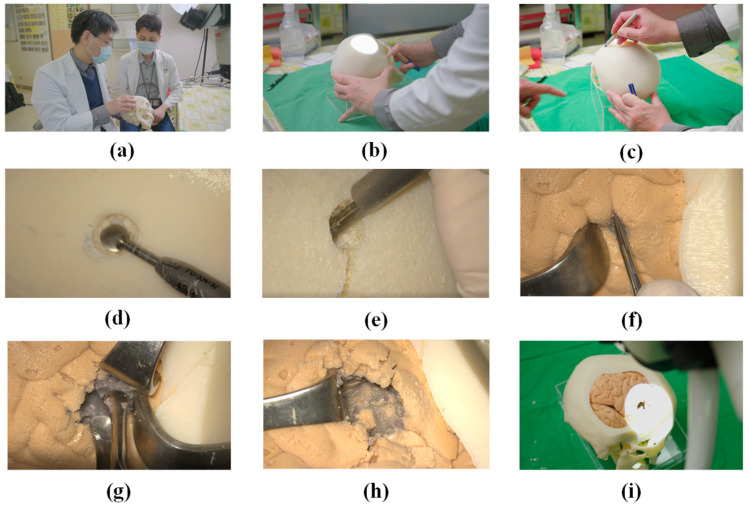
(**a**) Dr. Liu giving a neurosurgery trainee instructions on the use of the proposed simulator; (**b**) adjusting the angle and orientation of the head prior to commencing surgery; (**c**) marking the skull with three points to guide the trainee in sawing the top skull prior to craniotomy; (**d**,**e**) cutting off the top skull to expose the brain lobes; (**f**) making space between the brain lobes to locate the tumor; (**g**) using surgical forceps to remove the tumor; (**h**) removal of grey tumor tissue left a cavity within the lobe (infiltration scenario); (**i**) withdrawal of forceps and covering of skull to complete the operation.

## Data Availability

The data presented in this study are available on request from the corresponding author.

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
