# Peer review of "Advanced Manufacturing in the Fabrication of a Lifelike Brain Glioblastoma Simulator for the Training of Neurosurgeons"

_polymers, 2022, doi:10.3390/polym14061072_

Round 1

Reviewer 1 Report

In this work Pin-Chuan Chen and colleagues developed an artificial brain construct for simulating tumorectomy.  The authors made use of different additive manufacturing strategies to create an artificial construct that had the same tactile sensation to the native brain.  Although the formatting of this paper is not conventional, the development process is interesting and unique.  I believe this paper can eventually be published, but some issues need to be addressed.  Please consider the following:

1- The authors described the development of the materials used, along with the analysis of their mechanical properties.  I have some suggestions:

1.1- I believe the analysis of organ shrinkage should be changed.  Volume change, instead of height, should be evaluated due to the tridimensionality of the construct.

1.2- I believe figure 7 should be replaced.  I advise making a stress (MPa)-compression(%) curve and calculate the materials mechanically relevant properties: Young’s modulus, Maximum stress and Maximum compression.

1.3- The values inserted should be presented as Mean ± standard deviation, along with the n-value used for the experiment.

1.4- Consider also evaluating water content;

1.5. Consider presenting a tentative self/useful-life based on your results.

2- Please consider including the kind of patient this brain was modeled after;

4- I consider particularly interesting correlating different opinions from different neurosurgeons (senior and junior) to the materials/constructs mechanical properties, in order to best optimize the training model.  Can you provide a larger systematized study with more neurosurgeons on your model?

5- Consider optimizing the model for various stages of glioblastoma progression and provide a larger opinion study on the model.

Reviewer 2 Report

This paper is more like a case study and too commercial instead of a sound scientific research. I'm afraid it will not appeal too much interest from the general audience! A commercial or industrial applied journal might be more suitable for it.

Round 2

Reviewer 1 Report

In this work Pin-Chuan Chen and colleagues developed an artificial brain construct for simulating tumorectomy.  The authors made use of different additive manufacturing strategies to create an artificial construct that had the same tactile sensation to the native brain.  Although the formatting of this paper is not conventional, the development process is interesting and unique.  Most of my concerns have been addressed in the last round of reviews.  Before publication, I ask the authors to address the following issues:

  1. Consider adding the information provided in points 1.1, 1.4 and 1.5 to the manuscript.
  2. I understand now the issues with the determination of the composites mechanical properties. However, I found interesting the evaluation you performed. Please, consider adding the figure presented in point 1.2 to the SI section, and insert the information provided in point 1.2 in the manuscript.  Moreover, please provide the conditions for data acquisition and equipment used and insert it in the materials and methods section.
  3. Regarding the figure with the mechanical data, can you explain why some curves started with loads higher/lower than 0? If this was done to ease visualization, please consider changing the units in the x-axis to “a.u.”.
  4. Regarding point 2 of my last review, I apologize for not being clear. I wanted you to consider adding information regarding the patient used to model this construct: gender, age and concomitant pathologies. Please consider inserting this information for enriching your manuscript.
  5. Finally, please consider adding similar information regarding the future work and applications of this model (e.g. similar to the information provided in point 4 of my last review of the manuscript).

Reviewer 2 Report

It can be accepted!

Author Response

Thanks for your valuable comments.

Round 3

Reviewer 1 Report

In this work Pin-Chuan Chen and colleagues developed an artificial brain construct for simulating tumorectomy.  The authors made use of different additive manufacturing strategies to create an artificial construct that had the same tactile sensation to the native brain.  Although the formatting of this paper is not conventional, the development process is interesting and unique.  Most of my concerns have been addressed in the last round of reviews.  I recommend the publication of this manuscript.

This manuscript is a resubmission of an earlier submission. The following is a list of the peer review reports and author responses from that submission.

Round 1

Reviewer 1 Report

This paper is more like a commercial case study rather than a proper scientific work, as there is ZERO statistical experiments/data/analyses on the subject studied. I don't feel comfortable to have it published in a scientific journal.

Reviewer 2 Report

In this work Pin-Chuan Chen and colleagues developed an artificial brain construct for simulating tumorectomy.  The authors made use of different additive manufacturing strategies to create an artificial construct that had the same tactile sensation to the native brain.  Although the formatting of this paper is not conventional, the development process is interesting and unique.  I believe this paper can eventually be published, but not in its current format.  I am returning major suggestions with this decision, so to give the authors another change at describing the construct’s full potential.

1- The authors do not present a materials and methods section with the printing conditions/techniques, equipment used, material/filament preparation and respective suppliers.  Please include this.

2- How were the composites used in this work optimized?  Please describe the development rational.

3- The composites used should be fully characterized.  Apart from conventional chemical characterization (FTIR, TGA, DSC, XRD) the authors should focus on more specific assays, including 1) mechanical properties, 2) material stability/resilience for prolonged storage/manipulation, 3) morphological analysis of the construct using micro-CT.

4- I consider particularly interesting correlating different opinions from different neurosurgeons (senior and junior) to the materials/constructs mechanical properties, in order to best optimize the training model.

5- Consider optimizing the model for various stages of glioblastoma progression and provide a larger opinion study on the model.

6- Overall, I consider the results presented to be subjective and vague.  Please provide quantitative data to support your work.